

# Relationship between intended force and actual force: comparison between athletes and non-athletes

Alex Rizzato[1], Giovanni Cantarella[2], Elisa Basso[1], Antonio Paoli[1], Luca Rotundo[1], Patrizia Bisiacchi[2] and Giuseppe Marcolin[1]

[1] Department of Biomedical Sciences, University of Padua, Padua, Italy
[2] Department of General Psychology, University of Padua, Padua, Italy

Corresponding author
Giuseppe Marcolin,
giuseppe.marcolin@unipd.it

## ABSTRACT

This cross-sectional study aimed to investigate whether athletes (ATHL) and non-athletes (NON-ATHL) individuals had similar accuracy in matching intended to actual force during ballistic (BAL) and tonic (TON) isometric contractions. In this cross-sectional study, the subjects were divided into ATHL ($n = 20$; $22.4 \pm 2.3$ yrs; $73.2 \pm 15.7$ kg; $1.76 \pm 0.08$ m) and NON-ATHL ($n = 20$; $24.6 \pm 2.4$ yrs; $68.2 \pm 15.0$ kg; $1.73 \pm 0.1$ m) groups. The isometric quadriceps strength was measured with a load cell applied to a custom-built chair. For each condition, subjects performed at first three maximal voluntary isometric contractions (MVIC) as reference. Then, subjects had to match three intended force intensities expressed in percentage of the MVIC (*i.e.*, 25%, 50%, and 75%) without any external feedback. Subjects performed three trials for each force intensity. The accuracy (AC) was calculated as the absolute difference in percentage between the intended and the actual force. A Likert scale was administered for each trial to assess the subjective matching between the intended and the actual force. Statistical analysis showed that the ATHL group was more accurate ($p < 0.001$) than the NON-ATHL group. In contrast, the AC ($p < 0.001$) was lower when the force intensities increased independently from the group. Moreover, significantly higher AC ($p < 0.001$) and lower aggregate Likert scores ($p < 0.001$) were found in BAL than TON conditions. These results suggest that (i) sports practice could enhance muscle recruitment strategies by increasing the AC in the isometric task; (ii) differences between intended and actual force appeared to be intensity-dependent with lower AC at high force intensities; (iii) different control systems act in modulating BAL and TON contractions.

## INTRODUCTION

Intended force refers to muscle tension sensations experienced during contraction (*Jones & Hunter, 1983*). It results from a neuronal process influenced by two factors: the corollary discharge of the central motor command (*i.e.,* the motor signal informs the sensory system about the planned or implemented action) and afferent feedback from the working muscles (*Jones & Hunter, 1983*; *Pageaux, 2016*). Indeed, peripheral inputs from the neuromuscular

spindles and Golgi tendon organs provide information to higher centers about the in-place contraction (*Carson, Riek & Shahbazpour, 2002*; *Proske & Allen, 2019*). Moreover, the cerebellum ensures movement accuracy and sensory perception by communicating with the brain areas and sensory channels (*Bhanpuri, Okamura & Bastian, 2012*).

Historically, to the extent of standardizing the means of rating the perceived effort at specific muscular forces, scientific literature (*Cooper et al., 1979*; *Jones & Hunter, 1982*) focused on the relationship between actual and intended force in different muscle groups. The work by *Cooper et al. (1979)* suggested that a single motor performance, namely isometric or dynamic contraction, can be perceived with remarkable precision by both small (*e.g.*, adductor pollicis) and large (*e.g.*, quadriceps) muscle groups. However, a perceptual underestimation of force might occur in larger muscle groups, more commonly involved in gross movements, when lower levels of force are required to accomplish fine movements (*Jones & Hunter, 1982*).

More recently, the scientific interest moved to deepen the relationship between actual and intended force at different intensities of a maximal voluntary contraction. Indeed, in a motor task, the force intensity is influenced by the recruitment and firing rate of the motor unit and by the type of contraction (*i.e.*, ballistic or tonic) (*Miyamoto, Kizuka & Ono, 2020*). Ballistic contractions are characterized by high firing rates, brief contraction times, and high rates of force development (*Zehr & Sale, 1994*). Conversely, tonic contractions are sustained muscle contractions to keep persistent muscular tension and generate a maintained force (*Bagshaw, 1993*). Previous studies pointed out conflicting results on the force intensities and the subjects' accuracy in matching intended and actual force. In detail, West and colleagues showed that when subjects were asked to produce a specific force level during an isometric knee-extension task (*i.e.,* without maximal voluntary contraction reference), they were more accurate at lower intensities. In contrast, they overestimated the intended force in high-intensity contractions (*West et al., 2005*). Similarly, *Miyamoto, Kizuka & Ono (2020)* showed that the actual force did not accurately match the target contraction without visual feedback, with a more precise performance at high rather than low intensities. To the best of our knowledge, none of the above-mentioned studies considered the subjects' sports experience as a potential influencing variable in force modulation (*i.e.,* ballistic and tonic) at different intensities (*Baweja et al., 2009*; *Miyamoto, Kizuka & Ono, 2020*). Indeed, samples were scarcely described without considering that physical activity level and sports experiences could have affected the results. In this regard, athletes are likely to manage better physiological function, muscle contractions, and sports demands than their sedentary counterparts (*Thorstensson et al., 1977*; *Hagberg et al., 1988*). Moreover, team sports practice helps athletes more easily govern motor patterns than sedentary individuals due to their higher ability to perform motor tasks under stressful conditions in open-skill contexts (*Cortis et al., 2009*). Therefore, the primary outcome of this study was to investigate whether athletes and sedentary individuals had similar accuracy in matching the intended and the actual force during ballistic and tonic contractions in an isometric leg extension task. The secondary outcome was understanding at first glance if athletes had a better self-perception of their force production than non-athletes. In detail, hypothesizing that sports practice could positively influence subjects' ability to modulate

force, we expected that the group of athletes could better exert force at the required percentage of the maximal effort with a greater self-perception of the force production.

## MATERIALS & METHODS

### Subjects

Forty young, healthy subjects between 18 and 35 years old (age: $23.5 \pm 2.6$ yrs; mass: $70.7 \pm 15.3$ kg; height: $1.74 \pm 0.08$ m) were enrolled for the study. During the recruitment, subjects were divided into two groups: non-athletes (NON-ATHL; $n = 20$; $F = 13$; $24.6 \pm 2.4$ yrs; $68.2 \pm 15.0$ kg; $172.8 \pm 9.60$ m) and athletes (ATHL; $n = 20$; $F = 6$; $22.4 \pm 2.3$ yrs; $73.2 \pm 15.7$ kg; $1.76 \pm 0.08$ m). The following inclusion criteria were considered for the NON-ATHL group: (i) no history of competitive sports practice and (ii) no current practice of any physical exercise (*Thompson, Gordon & Pescatello, 2013*). For the ATHL group, inclusion criteria included (i) current sports practice and (ii) competing at a national level. Only subjects with no history of lower-limb injuries in the last year were eligible for inclusion in the NON-ATHL and ATHL groups. The athletes recruited practiced an individual ($n = 4$ track and field; $n = 1$ canoeing; $n = 1$ fencing; $n = 1$ gymnastics) or team ($n = 6$ basketball; $n = 3$ rugby; $n = 2$ volleyball; $n = 2$ soccer) sport discipline.

The experimental protocol received approval (HEC-DSB/05-21) from the Human Ethical Committee of the Department of Biomedical Sciences of the University of Padova and adhered to the principles of the Declaration of Helsinki. All the subjects involved in the study were informed about the methods and aims of the study, gave their written consent, and were free to renounce the study at any stage.

### Study design

We outlined a cross-sectional design in which subjects were asked to perform isometric leg-extension strength tests at different intensities expressed in percentage with respect to MVIC (*i.e.*, 25%, 50%, and 75%) without any feedback. Two different muscle contractions were studied: tonic (TON) and ballistic (BAL). In the TON condition, subjects reached the intended force level with a slow and maintained three-second muscle contraction. Conversely, in the BAL condition, the intended force had to be reached instantaneously. At a preliminary stage, subjects underwent the Waterloo Footedness-revised (WFQ-R) and Global Physical Activity (GPAQ) questionnaires to identify their dominant leg for the execution of the tests and to estimate their physical activity level, respectively. After each contraction, subjects were administered a Likert scale to assess their self-perceived accuracy in the strength task. Muscle contractions and the strength intensities within each type of contraction were randomized.

### Isometric lower-limb strength

We assessed the maximal isometric strength of the dominant quadriceps for each subject. To achieve this measure, we used a custom-built chair (Fig. 1A) and a load cell (MuscleLab™ 4100e; Ergotest Innovation, Porsgrunn, Norway). The load cell was placed three centimeters above the lateral malleolus of the dominant limb, perpendicular to the subject's shank. Subjects had to sit maintaining the knee angle at 90 deg. Straps on chest

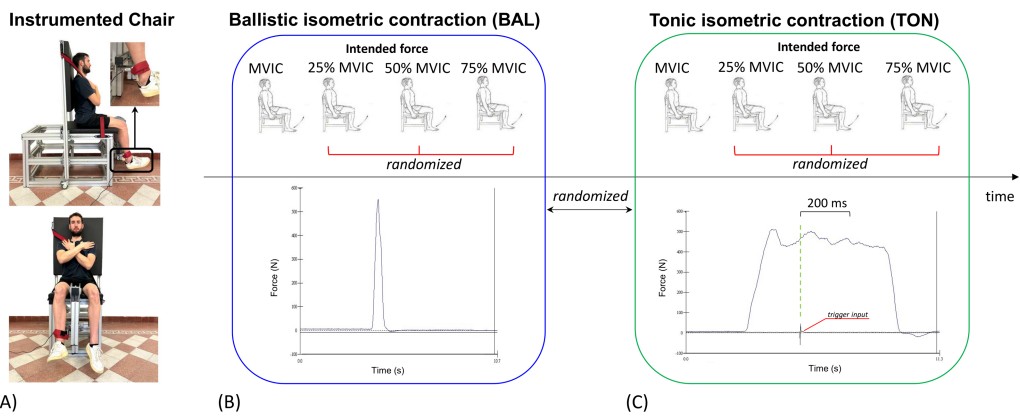

**Figure 1** **Graphical representation of the experimental design.** Custom-built chair instrumented with a uni-axial load cell (A); ballistic isometric condition with a representative force graph (B); tonic isometric condition with a representative graph of the force. The trigger input marked the beginning of the subjective matching between intended and actual forces, according to the subject's perception (C). MVIC: maximal voluntary isometric contraction.

and legs avoided any additional movement during the strength test. After a standardized warm-up (*i.e.,* a 10-min walk on a treadmill at 5 km/h and 10 repetitions of half-squat exercise), each subject was familiarized with the experimental setup by performing ten sub-maximal isometric quadriceps contractions with real-time visual feedback on the level of the exerted force. Specifically, the familiarization protocol consisted of five contractions at a submaximal self-selected intensity for both BAL and TON conditions. Then, after a 5-minute rest, subjects performed the test without any external feedback. Subjects were required to exert, for each of the two contractions (*i.e.,* BAL and TON), first the maximal voluntary isometric contraction (MVIC) and then the three submaximal intensities (*i.e.,* 25%, 50%, and 75% of the MVIC). Regardless of the condition and the intensities, three consecutive trials were performed for each intensity. A recovery of 40 s was adopted among trials, while a 5-minute rest occurred between the two contraction modalities. In the BAL condition, the subjects were asked to reach the required submaximal force intensity as fast as possible (Fig. 1B). Conversely, in the TON condition, the subjects had to hit an accelerometer (MuscleLab™ 4100e; Ergotest Innovation, Porsgrunn, Norway) to define the moment from which their actual force was supposed to correspond to the intended force (Fig. 1C). Load cell and accelerometer signals were synchronously recorded at 100 Hz with the dedicated software (MuscleLab™ 4100e; Ergotest Innovation, Porsgrunn, Norway).

## Questionnaires

*Waterloo Footedness Questionnaire-revised (WFQ-R).* The questionnaire (*Van Melick et al., 2017*) contains twelve items to address limb dominance specifically, and subjects marked a preference (*i.e.,* right or left limb) for the action indicated by each item. In detail, subjects were required to mark with a double sign (+ +) whether they had an absolute preference for a limb. Conversely, if they had a preference but could perform the action with both

limbs, they were asked to mark the preference with a single sign (+). Subjects marked both limbs with a single sign if they had no preference.

*Global Physical Activity Questionnaire (GPAQ-2).* Before trials, each subject was administered the GPAQ-2 to estimate the daily physical activity level (*Bull, Maslin & Armstrong, 2009*). With 16 items, GPAQ-2 covers the following physical activity components: intensity, duration, and frequency. It assesses the three domains in which physical activity is performed: (i) occupational physical activity, (ii) transport-related physical activity, and (iii) physical activity during discretionary or leisure time. The sum of the total Metabolic Equivalent (MET) of activity was computed for a typical week in each domain.

*Likert scale.* A single-item Likert scale was administered to subjects after each submaximal condition (*i.e.,* 25%, 50%, and 75% MVIC). It assesses the subjective matching between the intended force and the actual force. In detail, the single-item scale (*i.e.,* ''How confident are you in having correctly adhered to the required force?'') ranged from 1 (*not confident at all*) to 7 (*absolutely confident*).

## Data analysis

For the MVIC trials, the highest lower-limb peak force expressed in Newton in both TON and BAL conditions was considered among the three trials and normalized as a percentage of body mass expressed in Newtons. In the BAL contraction, the three peak values referred to each intended force condition were averaged. In the TON contractions, the force signal was averaged within a 200 ms time window starting from the trigger input of the accelerometer signal, similar to a previous study (*Miyamoto, Kizuka & Ono, 2020*). The actual force was always normalized to the body mass expressed in Newtons and reported as a percentage of the MVIC. The accuracy (AC) was calculated as the absolute difference between the intended and the actual force. In detail, the lower the AC values, the higher the accuracy, with a 0 score indicating the maximal accuracy. The aggregate Likert scores of the three trials for each force intensity (*i.e.,* 25%, 50%, and 75% of the MVIC) were considered for the statistical analysis.

## Statistical analysis

A priori power analysis (G*Power 3.1.9.2 software) showed that a total sample size of 36 participants and a medium effect size of 0.25 would provide a power of 0.8 with an alpha error probability of 0.01. The mean value among the three trials for both TON and BAL conditions were calculated for the MVIC, the submaximal intensity (*i.e.,* 25%, 50%, and 75% MVIC), and the AC. The Shapiro–Wilk test was used to check the data normality distribution of MET and MVIC values. Then, an unpaired sample $t$-test was used for baseline comparisons between groups (NON-ATHL *vs.* ATHL). Levene's test was used to check the equality of variances of AC values and aggregate Likert scores. Then, a three-way mixed-model analysis of variance (ANOVA) was used to investigate the main effect of force intensities (25%, 50%, and 75%), groups (NON-ATHL *vs.* ATHL), contractions (TON *vs.* BAL), and their interactions. For force intensities, in case of any statistically significant main effect or interaction, the Bonferroni post-hoc test was performed. The significance level was set at $p < 0.05$. JASP Software, version 0.16.3.0, was used for statistical analysis.

**Table 1  Baseline comparisons.**

|  | NON-ATHL Group | ATHL Group | *p* value |
|---|---|---|---|
| **MET** | 1408.00 ± 1333.96 | 7675.20 ± 4480.24 | $p < 0.001$ |
| **MVIC-BAL** (%BM) | 50.785 ± 25.784 | 77.38 ± 21.45 | $p < 0.001$ |
| **MVIC-TON** (%BM) | 69.34 ± 23.00 | 105.84 ± 19.48 | $p < 0.001$ |

Notes.
MVIC, maximal voluntary isometric contraction; BAL, ballistic contraction; TON, tonic contraction; NON-ATHL, non-athletes; ATHL, athletes; BM, Body Mass; MET, Metabolic Equivalent.
Data are presented as mean ± standard deviation.

**Table 2  Results of the actual force for all the intended force conditions (*i.e.*, 25%, 50%, and 75% MVIC).**

| | NON_ATHL Group | | ATHL Group | |
|---|---|---|---|---|
| Intended force | BAL (%MVIC) | TON (%MVIC) | BAL (%MVIC) | TON (%MVIC) |
| **25% MVIC** | 30.43 ± 13.65 | 16.05 ± 9.41 | 34.34 ± 11.09 | 23.76 ± 9.60 |
| **50% MVIC** | 44.03 ± 17.96 | 25.96 ± 11.35 | 47.09 ± 13.07 | 36.64 ± 14.92 |
| **75% MVIC** | 53.26 ± 19.12 | 35.08 ± 15.06 | 53.87 ± 14.63 | 46.45 ± 17.34 |

Notes.
MVIC, maximal voluntary isometric contraction; BAL, ballistic contraction; TON, tonic contraction; NON-ATHL, non-athletes; ATHL, athletes.
Data are presented as mean ± standard deviation.

## RESULTS

All the subjects completed the study. Results of the WFQ-R questionnaire showed that thirty-nine subjects were right dominant and one was left dominant. Unpaired *t*-test for baseline comparisons showed statistically significant differences between NON-ATHL and ATHL groups for MET and MVIC of both BAL and TON contractions (Table 1).

Strength values expressed at 25%, 50%, and 75% MVIC in both BAL and TON contractions are presented in Table 2.

The AC results are presented in Fig. 2. The three-way ANOVA analysis showed a significant main effect of contractions ($p < 0.001$; $F = 19.83$; $\eta_p^2 = 0.34$), force intensities ($p < 0.001$; $F = 63.41$; $\eta_p^2 = 0.63$), and an interaction between them ($p < 0.001$; $F = 14.10$; $\eta_p^2 = 0.27$). The Bonferroni post hoc comparisons are shown in Fig. 2 and highlight the significant differences among force intensities. A significant main effect of the group ($p < 0.05$; $F = 4.58$; $\eta_p^2 = 0.10$) was also observed. The between-group (ATHL *vs.* NON-ATHL) differences in percentage showed a greater accuracy of the ATHL: (i) BAL contraction (25% MVIC: 3.79%, 50% MVIC: 33.54%, and 75% MVIC: 9.60%) and (ii) TON contraction (25% MVIC: 27.98%, 50% MVIC: 30.36%, and 75% MVIC: 26.76%).

The results for the aggregate Likert scores in the 25%, 50%, and 75% MVIC in both BAL and TON contractions are presented in Fig. 3. The three-way ANOVA analysis showed a significant main effect of contractions ($p < 0.001$; $F = 48.85$; $\eta_p^2 = 0.56$) and a significant interaction between force intensities and groups ($p < 0.05$; $F = 3.49$; $\eta_p^2 = 0.08$).

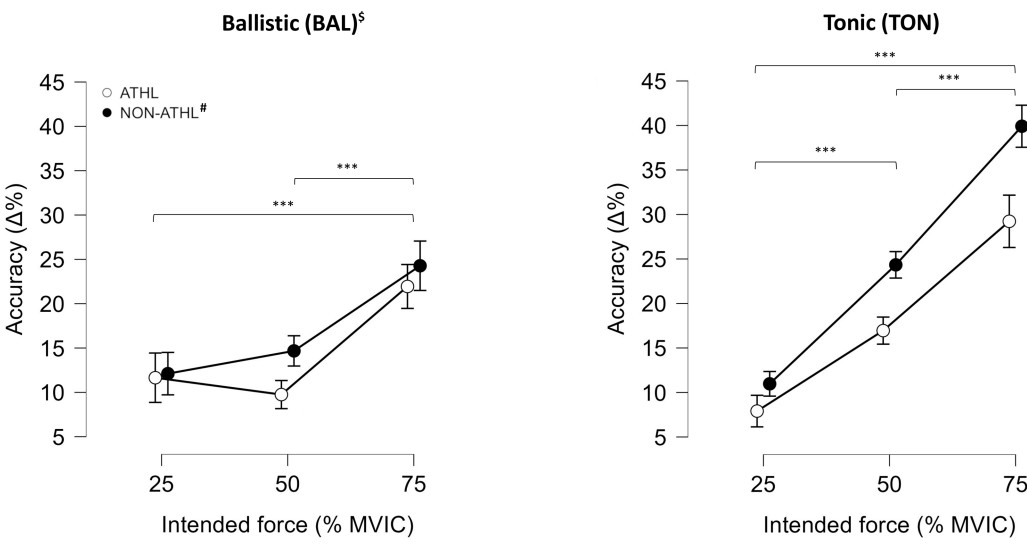

**Figure 2 Isometric strength result.** Accuracy (AC) in athletes (ATHL) and non-athletes (NON-ATHL) individuals during ballistic and tonic contraction at the different force intensities (25%, 50%, and 75% MVIC). Data are presented as mean ± standard error. *** ($p < 0.001$). # Statistically significant groups effect ($p < 0.05$). $ Statistically significant contractions effect ($p < 0.001$).

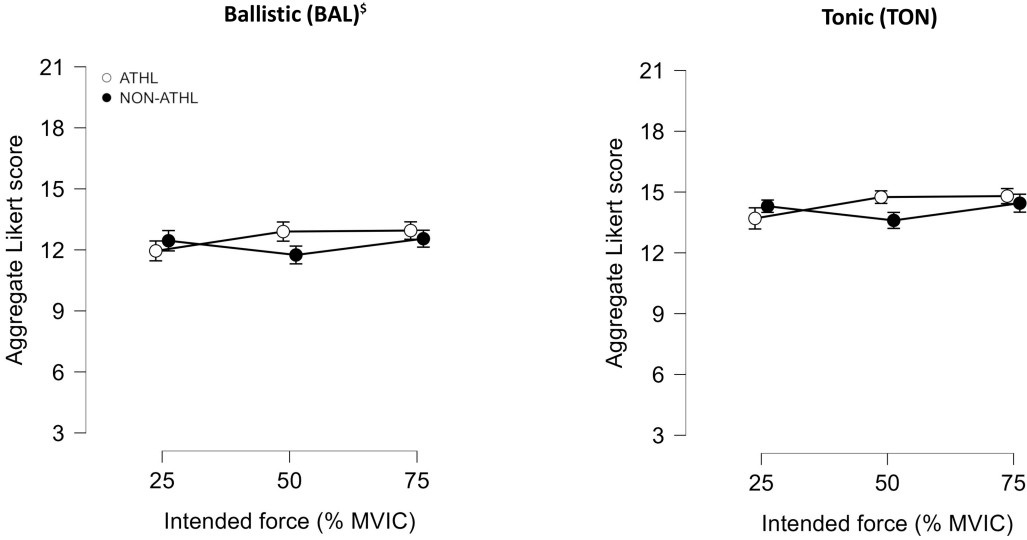

**Figure 3 Results of the aggregate Likert scores.** Aggregate Likert scores between actual and intended force in athletes (ATHL) and non-athletes (NON-ATHL) individuals during ballistic and tonic contraction at different force intensities (25%, 50%, and 75% MVIC). Data are presented as mean ± standard error. $ Statistically significant contractions effect ($p < 0.001$).

## DISCUSSION

This study aimed to investigate the relationship between intended and actual force, comparing athletes and sedentary individuals during BAL and TON contractions. The main finding confirmed our hypothesis, showing that the ATHL group could exert force at the required MVIC percentage more accurately than the NON-ATHL group, independently of the type of contraction. Since the recruitment and firing rate of the motor units and the contraction type could affect both force production (*Enoka & Fuglevand, 2001*) and the relationship between intended and actual force (*Miyamoto, Kizuka & Ono, 2020*), we can assume that sports practice could enhance muscle recruitment strategies (*Enoka & Fuglevand, 2001*) and the ability to modulate force.

Similarly, the higher levels of physical activity and training of ATHL account for the higher MVIC values. Therefore, the baseline results supported the hypothesis that the subjects' sports practice could be an influencing variable in force production and claimed for group division when investigating the relationship between intended and actual force. Regarding the force values recorded in the present study, it is interesting to note that, under most conditions and independently from the group, the actual force tended to overestimate the intended force (Table 2). Indeed, this underproduction of force was in line with previous results performed on isometric knee extension (*Cooper et al., 1979*), isokinetic knee extension (*Jackson et al., 2006*), isometric elbow flexion and extension (*John, Liu & Gregory, 2009*), and isometric thumb adduction (*Cooper et al., 1979*). Although the causes of errors in estimating muscular effort have yet to be completely understood, some possible speculations to account for low levels of accuracy could be addressed. Indeed, it has been theorized that during voluntary motor task performance, descending motor command signals include physiological noise derived from neuronal and synaptic action potential discharges, leading to muscle force production variability (*Harris & Wolpert, 1998*). In detail, neuronal noise increases with the magnitude of the motor command (*i.e.,* the intensity of a motor task) (*John, Liu & Gregory, 2009*). Moreover, transient post-contractile potentiation of spinal reflex pathways may summate with previously set motor commands, producing errors in effort perception (*Hutton, Enoka & Suzuki, 1984*).

Even though the actual force most undershot the intended force, the deviation from the actual force was less marked in the ATHL than in the NON-ATHL group. It is generally accepted that force modulation is accomplished by combining the rate coding of individual motor units and the recruitment of more or fewer motor units (*Clamann, 1993*). Moreover, muscle adaptations (*e.g.*, following training) at both neural and morphological levels underpin the enhanced ability of force regulation in skeletal muscle (*Folland & Williams, 2007*; *Carroll et al., 2009*). Thus, chronic exposure to training and exercises with repetitive movements against loads for a long time, as athletes regularly do, could have substantially increased their ability to modulate force (*Maden-Wilkinson et al., 2020*).

In both NON-ATHL and ATHL groups, a better accuracy was observed at lower (*i.e.,* 25% MVIC) than higher (50% and 75% MVIC) force intensities. Thus, our finding suggested that differences between intended and actual force appeared to be intensity-dependent. Previous studies showed that the presence of an anchoring process (*i.e.,* MVIC as reference),

as in our study, addressed results in different directions. *West et al. (2005)* reported that when the experimental task was performed before the MVIC, the actual force matched the intended force at the intermediate level (*i.e.,* 50% MVIC) while without the anchoring process, the most accurate match occurred at 25% MVIC. In contrast with our results, other studies on precision grip (*Kumar, Narayan & Chouinard, 1997*), power grip (*Kumar, Narayan & Chouinard, 1997*), and chest press (*Jackson & Dishman, 2000*) showed that the actual force matched the intended force at intermediate level (*i.e.,* 40–50% MVC).

In our study, the matching between the intended and actual force was lower at higher intensities (*i.e.*, 50% and 75% MVIC). Thus, the intended force resulted in a more marked perceptual overestimation in these conditions. Hence, our results suggested that a perceptual mistake occurred at higher force intensities even with previous knowledge of the maximal effort (*i.e.,* the MVIC) as an anchoring process. Even though our data cannot explain why subjects produced the most accurate match at the lower force intensity (25% MVIC), two possible speculations may be acknowledged. First, *West et al. (2005)* explained that subjects subconsciously under-produced force at higher intensities as a protective mechanism. Indeed, the perceptual overestimation occurred to maintain a reserve capacity and prevent mechanical and metabolic damages, which may arise in further force generation. Second, lower force percentages are used in everyday contexts, most for simple motor tasks (*e.g.*, walking or getting up from a chair). Therefore, subjects could have performed a better perceptual estimation because of the proximity to the levels of force they are accustomed to, regardless of the group.

Our findings highlighted that the relationship between intended and actual force was more accurate in the BAL than in the TON condition. Since the BAL contraction occurred rapidly and without afferent sensory feedbacks (*Vincken, Gielen & Van der Gon, 1984*), it was mostly controlled by central predetermined regulatory mechanisms through a feedforward strategy (*Hanneton et al., 1997*). Conversely, TON contraction had a longer duration, and during this time, information from sensory receptors could interfere with the ongoing task (*Desmurget & Grafton, 2000*). Indeed, feedback control mechanisms are predominant during the skill acquisition process and essential for constructing a newly learned movement (*Seidler, Noll & Thiers, 2004*). In our case, the isometric contraction at leg extension represented a basic motor behavior, where corrective feedback used in complex motor skills was mainly unnecessary. In this regard, *Lohse, Sherwood & Healy (2011)* found that an internal focus of attention (*i.e.,* focusing on muscle contraction) decreased accuracy in basic motor behavior (*i.e.,* 30% MVIC) compared to an external focus of attention (*i.e.,* focusing on pushing over the force platform). Similarly, subjects in the longer TON contraction could have internally directed their attention on muscles, calling into play counterproductive corrective feedback. This hypothesis could account for the worse accuracy in the TON than BAL contraction, where feedforward mechanisms were revealed to be more effective.

Finally, subjects' self-perception, deepened through the Likert scale rating at the end of each contraction, highlighted no differences between groups. This result was unexpected since the ATHL group was more accurate in force production, implying a greater awareness of force modulation. However, the motor task studied in this research was a simple one.

Thus, we can speculate that a more sport-oriented task could have differentiated the self-confidence performance between athletes and non-athletes. Moreover, a main effect of the contraction type was observed in the Likert aggregate scores, with a greater self-confidence towards the TON against the BAL contraction. Since the BAL contraction was the most accurate, we can speculate that the longer time the subjects had to modulate the force led to a perceptual mistake in their self-perceived accuracy in the TON condition. Nonetheless, further research should deepen the self-perception of force production, considering also the between-trial variability of the Likert scores.

The present study has some potential limitations to be acknowledged. Although both groups included male and female subjects, it was impossible to perform a sex analysis because the relatively small number of males and females in both groups would not have allowed sufficient statistical power. Then, athletes practiced different sports. Thus, recruiting athletes from similar sports with the predominant use of lower limbs could have enhanced the differences with non-athlete peers. Moreover, a dynamic or sport-oriented task could have improved accuracy and self-confidence, being closer to the infield practice.

## CONCLUSIONS

The present study expanded the previous results on the relationship between the intended and the actual force by investigating whether sports practice could be an influencing factor. Our findings showed that athletes could exert force at the required MVIC percentage more accurately than non-athletes, independently of the type of contraction. Since the accuracy between intended and actual forces seems to decrease as a function of the increment of the force intensity, our results suggested that coaches and practitioners should focus on training the force accuracy at high intensity. In most situations, sports actions require fast movements and a high level of strength; thus, the accuracy at a high force level in ballistic movements is supposed to be an important factor in achieving performance. Indeed, repetitive training should aim to improve the accuracy of force production at high force levels. Differences in accuracy between ballistic and tonic contraction suggest different control mechanisms. Further work is needed to explore these hypotheses and to deepen the perceptual overestimation detected at the higher isometric force intensities.

## ACKNOWLEDGEMENTS

The authors would like to thank all the subjects who voluntarily participated in the study.

### Funding

The authors received no funding for this work.

### Competing Interests

Giuseppe Marcolin is an Academic Editor for PeerJ.

## Author Contributions

- Alex Rizzato conceived and designed the experiments, analyzed the data, authored or reviewed drafts of the article, and approved the final draft.
- Giovanni Cantarella performed the experiments, analyzed the data, prepared figures and/or tables, and approved the final draft.
- Elisa Basso performed the experiments, analyzed the data, prepared figures and/or tables, and approved the final draft.
- Antonio Paoli conceived and designed the experiments, authored or reviewed drafts of the article, and approved the final draft.
- Luca Rotundo performed the experiments, prepared figures and/or tables, and approved the final draft.
- Patrizia Bisiacchi conceived and designed the experiments, authored or reviewed drafts of the article, and approved the final draft.
- Giuseppe Marcolin conceived and designed the experiments, authored or reviewed drafts of the article, and approved the final draft.

## Human Ethics

The following information was supplied relating to ethical approvals (*i.e.*, approving body and any reference numbers):

The name is: Human Ethical Committee. Department of Biomedical Sciences, University of Padua (HEC-DSB/05-21).

## Data Availability

The raw data is available in the Supplemental Files.

## Supplemental Information

Supplemental information for this article can be found online at http://dx.doi.org/10.7717/peerj.17156#supplemental-information.

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
