# Peer review of "Relationship between intended force and actual force: comparison between athletes and non-athletes"

_PeerJ, doi:10.7717/peerj.17156_

## Round 0.1 · original submission · Major Revisions

All reviewers noted minor and major comments to the manuscript. Most of them are focusing on the statistical/data analysis, which needs to be justified by the authors. I encourage the authors to address all reviewer comments and provide information or change the respective paragraphs if required.

·

Basic reporting

no comment

Experimental design

no comment

Validity of the findings

no comment

Additional comments

General comments:
The present study was designed to examine the association between the intended and the actual knee extensor force in athletes and sedentary participants. Therefore, isometric ballistic and tonic contractions were performed after maximum voluntary isometric contractions of the quadriceps muscle. The accuracy was quantified as percentage difference and was subjectively rated on a Likert scale. Data were analyzed using Shapiro-Wilk tests, Student’s T tests, Mauchly’s tests and three-way ANOVAs. Based on their findings, authors concluded that the athletes exerted submaximal force levels more accurately as the sedentary participants. This was more pronounced for the ballistic contractions.

The biggest issue I see with the work presented is its complex factorial design. The results of a three-way mixed ANOVA are very difficult to interpret. In this regard profile plots are helpful and recommended to uncover the complex relationships (cf. Ch. 19 in Keppel 1991). In case of significant interaction effects the main effects are hardly interpretable. Profile plots indicate whether main effects are interpretable (cf. Ch. 7 in Hair et al. 2010).


Detailed comments:
Abstract:
Lines 35 and 36: Please consider rephrasing.
Line 38: Please use the abbreviation for “accuracy”.
Line 39: Please use the abbreviation for “accuracy”.
Introduction:
Lines 75 and 80: Same citation with different years. Please correct.
Methods:
Line 99: Please replace “basket” with “basketball”.
Lines 126 and 128: The participants were familiarized with ten submaximal isometric contractions. At which intensities the familiarization session occurred? Did all participants use a similar familiarization protocol? Were both contraction types included?
Line 132: Is there any rationale behind the used recovery period of 40 seconds?
Line 167: Why did you considered the sum and not the average of the Likert scores reported?
Lines 174 and 175: What about the distribution of the accuracy data?
Lines 179 and 182: Please specify the post-hoc test.
Results:
The authors may wish to consider sub-headings to better structure the results.
Lines 184 and 186: Personally, I would be interested in whether the levels of MET or MVIC are related to the degree of inaccuracy in ballistic and tonic contractions.
Lines 195 and 197: This result is very important. The authors may wish to extent the reporting as well as the discussion of the group differences.
Discussion:
Line 227: Did your data confirm this theory? Did the results of the trials gradually decrease?
Figure 3: Please add the minimum and maximum possible values [3-21] at the y axis.

References used for review:
Hair JF, Jr., Black WC, Babin BJ, Anderson RE (2010) Multivariate data analysis. Pearson Prentice Hall
Keppel G (1991) Design and analysis: a researcher's handbook. Prentice-Hall, Upper Saddle River

·

Basic reporting

Language
The English language in this manuscript could be improved. I suggest to ask a colleague who is a fluent speaker or to consult a professional service.
Examples:
Please rewrite the sentence starting in line 25f because it entails wrong grammar.
L38: Could you please clarify and/or rewrite what is meant by: “protective mechanisms or proximity to the levels of daily living forces could account the decrement of accuracy at high force intensities”.
L301: Please check grammar in this sentence and rephrase.

Experimental design

L29f: Three trials for each level or three trials in total (one for each level). This needs to be clarified in the abstract.

L100: The term “competitive” level is not very precise and maybe understood differently by different researchers (depending on the country). Can you specify this?

Information on subject recruitment strategies is missing. Were subjects retrospectively allocated to the two groups or did you specially “search” for the two groups. Please add recruitment information and applied inclusion/exclusion criteria (if used).

The questionnaire WFQ-R was used to assess limb dominance, however corresponding data are not presented in the Results.

l161f: Could you please add more information on the normalisation procedure, especially the units and normalisation to the body mass. Please also provide references on this approach (if available)
In this context, more baseline data in the Results section would be helpful to assess for potential confounding factors (age, weight etc.)

Information on force recordings is missing (software, filters used, sampling rate etc.). Please add.

l163: Why did you choose to analyse 200ms? Did you analyse other time windows? Please also use references to back up your approach.

L165: How was the difference in percentage between the intended and actual force calculated (intended – actual or vice versa?). This is crucial for interpreting Fig. 2. Please add this information.

l177: delete significant. You are not testing significant effects, you are testing if there are certain significant effects/interactions.

Validity of the findings

L35: It is not clear from the sentence which comparison was calculated. What was significantly greater?. BAL > TON? Please clarify.

l193: You did not mention Bonferroni-Holm post hoc comparisons in the methods section. Hence, it is not clear which post-hoc comparisons (between groups and/or intensities) were performed (this is also not clear in the figure). Please revise.

l196f: please clarify what % values are presented (differences between ATL and SED?). These values do not match those presented in Fig. 1. and Tab2.

Please also report the non-significant effects and interactions of the ANOVA and post-hoc comparisons.

Table 1: The MET minutes per week appear to be very high, especially the 1400 MET per week in the SED considering that the WHO recommends at least 600 metabolic equivalent minutes (Met minutes) of physical activity. Could you comment on that?

In this context, it would also be very helpful to know your definition of “sedentary”. Is this only based on MET minutes per week? How was this considered during subject recruiting (see previous comment).

Table 2: The differences in AC between the two groups appear to be more pronounced in tonic conditions and less so in ballistic condition. Were these differences statistically different? Please provide statistical results to demonstrate that AC differences between groups was independent of the contraction type (as stated in l207).

Fig. 3: The presented values are not clear. In the methods section, you report the Likert scale to be between 1 and 7 while in Fig. 3 and raw data much higher values are reported (up to 20) (see also comment on raw data).

L217 and L230: According to Table 2, actual force is higher in the 25% MVC condition, but not in the other two conditions, so the word globally is misleading. Please rephrase your conclusion. This is actually a very interesting finding because direction of differences between intended and actual force appears to be intensity-dependent.

L249: The word “associated” is misleading considering your statistical analysis. Please rephrase.

Could it be that the better AC in the ATL group is not generally linked to sport practice but specifically to the individual MVC. Did you check for an association of MVC and AC in the task?

Raw data
Thanks for providing the raw data, however when opening the excel file of the accuracy data, the raw data appear in the wrong format (i.e. in dates rather than numbers). This is probably a formatting issue. Please ensure that the data are reported correctly when opening the file.

In addition, I don´t understand why Likert scale data presented in the results section and in the raw data appear to range from 0 to 20 instead of 1 to 7 as described in the Methods section. Please clarify.

Reviewer 3 ·

Basic reporting

• In general language usage often contains intensions in contrast to definite findings (example from abstract: Moreover, significantly greater AC (p< 0.001) and subjective matching were found in BAL and TON contractions, respectively). This occurs repeatedly throughout the manuscript. Please check and correct.
• Further point for language use: although other articles already used "ballistic" to describe short contractions this, at least at first sight is confusing. Ballistic describes physical "phenomena of projectiles in flight". As I don't have a solution for this problem and other articles already used this term to describe similar experiments this is more a statement than an advice for change, but probably a short explanation or introduction to the matter would be nice.
• Provided background is appropriate
• Structure conforms PeerJ standards
• Figures are of appropriate quality, but at least partly not well labelled (see validity of findings).
Figure 2: two times p<0.001 ?, further: for all between group comparisons significant differences are labelled, which, according to recalculation cannot be replicated.
• Raw data are supplied, but only for AC values and Likert scale values. Raw force data are missing.

Experimental design

• Research is original, question is well defined. The knowledge gap to be filled is defined
• Investigation standard is appropriate and reported with sufficient detail to be replicated
• Group of SED and ATL consist of considerable different ratios of female subjects. This could be a possible bias for the results, but is not considered in the results and discussion – this calls for specific treatment of this matter (maybe in gender-specific analyses). Also, practised sports disciplines varied a lot.
• In the methods it is stated that the load call was placed 3 cm above lateral malleolus, but in figure 1 it looks like the load cell was located much more proximal. So the question is, if positioning of the load cell was really standardized the way the authors reported?
• Why an accelerometer (L135) was used for the trigger (provide rationale)?
• In the statistical analysis chapter there is no hint towards the used Bonferroni-Holm correction of the statistical results (see figures 2&3). This has to be introduced here. Further: how many comparisons were considered as the start correction factor?
• You seem to mainly adhere to significance levels, but also provide effect sizes (which is good). By looking at the respective ES levels some of the results move from significance towards non-relevant. This should be reported adequately.
• Lines 195-197: where are the data for these tests?

Validity of the findings

• Findings are not valid (according to re- calculation): already reported above: please recalculate – see below
• Provided Accuracy database: how was the relative error calculated? All values - except one - are positive numbers, but at 25% MVIC BAL in both groups the actual forces were in average larger than the intended ones. Not so for the other force levels. So it seems, that absolute deviations were considered for the accuracy calculation. But why there is one negative value? This questions the statement of general overestimation of actual force level. Please explain.
• Figure 3: why do you use the sum of the three singe trials? By treating the data this way, between trial variations cannot be detected, but may have occurred.
• Table 1: As far as I know the MET has no unit. Please explain and provide the respective reference
• As many open questions about the results and their interpretation remain (see experimental design) the discussion needs a critical and considerable update.

---

## Round 0.2 · Major Revisions

While reviewer 1 and 2 are mainly satisfied with the response letter of the authors, reviewer 3 still raises significant concerns which need to be addressed.

·

Basic reporting

No comments

Experimental design

No comments

Validity of the findings

No comments

Additional comments

The authors are to be congratulated for their effort in addressing my comments. I have no further comments. All points have been addressed.

·

Basic reporting

no comment

Experimental design

no comment

Validity of the findings

L198: A sentence on the main findings reported in Table 2 would be helpful.

Reviewer 3 ·

Basic reporting

n.a.

Experimental design

n.a.

Validity of the findings

n.a.

Additional comments

The authors have submitted a revised manuscript, but still there remain some questions. In their rebuttal only some statements like “we enlarged the description on this…” or something similar are provided, but not the exact position and the made changes. To adequately respond to the reviewers comments I would expect exact and detailed answers.
Moreover, the made changes were not always satisfactory to me.

Detailed comments.
Figure 2 and Table 3, and also figure 3 and table 4 contain identical data, but in a different way to present them. It is not only unusual, but prohibited to double display identical data in scientific articles. This has to be changed.

Figure 2
The provided statistics seem to test differences between force levels, but no difference between groups were provided.

Figure2 & 3
In the original version SD values were provided, here SE values were used. Why?

At the end of the results section (L207 in the pdf) you report a value of 33.54% deviation in the BAL condition, whereas the two other values were 3.78% and 9.60%, respectively. Please check.

Remaining questions from first review
I asked for gender-specific analyses, but got a peculiar answer which is far from being a serious reaction to my comment. In your NON-ATHL group there were six female subjects and in the ATHL group there were 13 female subjects. This sums up to 19 females - almost half of the total group size. You could just use gender as group variable to check, if gender has impact on the results (independent of group assignment to NON-ATHL and ATHL).
This does not necessarily has to be included in the results, but should be checked.

In my review I asked one question about the applied Bonferroni-Holm correction. Although you answered to this comment, you did not provide the required information. To my knowledge, the Bonferroni-Holm correction is an adaptive method for numerous p-values to avoid falsely detected significances (so called alpha statistical error). If you really used this correction - what exactly was the starting correction multiplier? Further: why Bonferroni-Holm, what were the criteria?

In my fist review I asked about effect sizes and their implications for the calculated significance levels. Although you stated to have weighted the findings in the discussion, due to the lack of details where to find them I could not recognize such changes.

In my first review I asked for the rationale to use summative values for the three trials, because by providing such data no between trial variances could be detected. In your answer you detailed the similarity between sum and mean. Sorry – this is not an adequate answer to my question.

---

## Round 0.3 · accepted · Accept

All reviewer comments were implemented.

Reviewer 3 ·

Basic reporting

NA

Experimental design

NA

Validity of the findings

NA

Additional comments

NA